# Bacterial assemblages on eggs reflect nesting strategies in wetland-associated birds

Wouter F. D. van Dongen[1,¤a], Hanja B. Brandl[1,¤b]*, Alžbeta Darolová[2], Ján Krištofík[2], Herbert Hoi[1]

1 Konrad Lorenz Institute of Ethology, Department of Integrative Biology and Evolution, University of Veterinary Medicine Vienna, Vienna, Austria, 2 Institute of Zoology, Slovak Academy of Sciences, Bratislava, Slovakia

☉ These authors contributed equally to this work.
¤a Current address: College of Sport, Health and Engineering, Victoria University, Victoria, Australia
¤b Current address: Department of Collective Behaviour, Max Planck Institute of Animal Behavior, Konstanz, Germany and Centre for the Advanced Study of Collective Behaviour, University of Konstanz, Konstanz, Germany
* hbrandl@ab.mpg.de

## Abstract

Birds host diverse bacterial assemblages, which play a critical role in individual health, but which can also lead to disease or mortality. It is therefore important for developing embryos to acquire appropriate bacterial communities from maternal (vertical transmission) and environmental (horizontal transmission) sources. Eggshell bacterial assemblages are acquired before and after oviposition, and are shaped by external factors, including habitat, nesting material and parental incubation. Understanding the source of eggshell bacteria is important, because eggshell penetration of horizontally-transmitted bacteria can affect embryonic health. Most research on eggshell-associated bacteria has occurred on 'dry-nesting' terrestrial birds. However, little is known on bacterial acquisition in waterbirds, particularly in nests where eggs are in direct contact with water. Moist environments favour bacterial growth and wet-nesting species are therefore expected to have higher bacterial loads. To date, no study has focussed on contrasting the abundance and diversity of eggshell bacterial assemblages in wet- and dry-nesting species. We used both bacterial culture and genetic techniques (automated ribosomal intergenic spacer analysis) to document the bacterial load and assemblage structure of eggshell-associated bacteria in both wet- and dry-nesting wetland-associated bird species. Bacterial loads were several orders of magnitude greater on eggs of wet-nesting species and bacterial assemblages tended to cluster by nesting strategy. These findings suggest a possible association of eggshell-associated bacteria with nesting strategies in these species. Further research is, however, required to confirm these patterns, incorporating more comprehensive sampling and utilising more advanced genetic approaches. Overall, our findings highlight a promising direction for future research into the association between nesting in moist environments and eggshell-associated bacteria, as well

**Data availability statement:** All relevant data are within the manuscript and its Supporting information files (S7 Datafile).

**Funding:** This work was supported by the Scientific Grant Agency of the Ministry of Education, Science, Research and Sport of the Slovak Republic and the Slovak Academy of Sciences, VEGA (project number 2/0077/11).

**Competing interests:** The authors have declared that no competing interests exist.

as the potential for antimicrobial adaptations that may characterise the eggshells of these species.

## Introduction

Bacteria play a central role in the biology of animals, occurring both internally [e.g., gastrointestinal tracts; 1,2] and externally [e.g., feathers and skin; 3,4] of their hosts. Although many bacteria are commensal, others are crucial in key processes such as immune function and nutrition [5,6] or have pathogenic properties [e.g., 2,7,8]. The bacteria that animals acquire during early development can therefore play an important role in determining their subsequent fitness.

Bacterial acquisition can occur both horizontally between an individual and its environment, and vertically from the individual's mother. In birds, vertical transmission of egg-associated bacteria occurs in the oviduct prior to eggshell formation, while horizontal transmission can occur via trans-shell penetration after oviposition [9]. The bacteria acquired by embryos prior to hatching can be beneficial, but transmission of pathogenic bacteria can result in embryonic mortality [7]. Several mechanisms therefore exist that influence the bacterial composition of bird eggshells, and the subsequent probability of embryonic infection by pathogens. These mechanisms include behavioural strategies [e.g., egg incubation or lining nests with nesting material with antimicrobial properties; 10], chemical [e.g., maternal investment into antimicrobial substances such as lysozyme and ovotransferrin; 11,12] or structural [e.g., nanostructural architure of the eggshell; 13].

The composition of eggshell bacterial assemblages is also known to be shaped by the ecology and life history of birds, including habitat and nest-type. For example, the eggshells of rural birds may host different bacteria to urban birds of the same species [14], while cavity nests are known to harbour more bacteria than open nests [15]. In wetland-associated birds, wet or humid nest conditions can favour bacterial growth and therefore present an additional challenge to controlling microbial growth on eggshells [16]. Darolová et al. [17] showed that wetland-nesting birds have greater bacterial loads on eggshells than terrestrial species, but not a higher maternal investment in antimicrobial substances in eggs. However, different nesting strategies are pursued within wetland-nesting species, with some constructing 'wet-nests' (eggs are in constant contact with water or soaked nesting material) and others constructing 'dry-nests' (eggs do not come into contact with the water or moist nesting material). This presents an informative opportunity to explore how nesting strategies within habitats influence the microbial composition of eggs. It remains unknown whether the eggs of wet-nesting birds have higher bacterial loads, and harbour different bacterial assemblages, to dry-nesting species among wetland birds. Documenting these differences is a crucial step in understanding the antimicrobial mechanisms involved in controlling bacterial growth in environments that promote bacterial proliferation.

Here we use both bacterial culture and genetic techniques to describe variation in bacterial assemblages associated with eggshell surfaces in thirteen wetland bird species that display either wet-nesting or dry-nesting reproductive strategies. We first use bacterial culture to investigate whether loads of known bacterial taxa are higher on

the eggs of wet-nesting species. Since only a small fraction of bacteria can be cultured, we also used genetic fingerprinting of bacterial samples and Bray-Curtis community similarity indices for a broader exploration of whether bacterial communities cluster by nesting strategy.

## Materials and methods

### Sampling locations

This study was conducted in May to July 2015, which corresponds to the breeding season of the study species. The research permit for this study was provided by the Ministry of Environment of the Slovak Republic (permit number 2971/2015–2.3). Sampling occurred in various lentic waterbodies in western Slovakia, including two fish ponds (Veľké Blahovo [70 ha]: 48º03′04″ N, 17º35′37″ E and Jakubov [24 ha]: 48º24′ 42″ N, 16º57′56″ E) and two water reservoirs (Vráble [36 ha]: 48º15′25″ N, 18º17′40″ E and Budkovany [3.6 ha]: 48º46′16″ N, 17º10′57″ E). These waterbodies are partly covered with marsh stands consisting of common reed (*Phragmites australis*), lesser reed mace (*Typha angustifolia*), great reed mace (*T. latifolia*), and *Carex* spp. (Cyperacae).

### Study species and nesting strategies

During the breeding season, we slowly walked through the study sites and searched for nests containing eggs. Nests were found via observations of the behaviour of adult birds and using our previous experience of each species. To investigate the influence of nesting strategy on eggshell bacteria, we classified each of the thirteen study species as wet-nesters or dry-nesters. For examples of wet and dry nests, refer to S1 Fig. The wet-nest category included two closely related species: the great-crested grebe (*Podiceps cristatus;* n = 43 nests) and the little grebe (*Tachybaptus ruficollis*; n = 8 nests). In these species, the eggs in the nest lie in the water or in soaked nesting material. In addition, the eggs are covered with wet plant material when parents leave the nest [18]. All other species were classified in the broad category of dry-nesters, as eggs were located on a dry inner nest layer. This category included the mallard (*Anas platyrhynchos*; n = 2 nests), pochard (*Aythya ferina*; n = 5 nests), red-crested pochard (*Netta rufina*; n = 1 nest), greylag goose (*Anser anser*; n = 3 nests), mute swan (*Cygnus olor*; n = 4 nests), common coot (*Fulica atra;* n = 28 nests), moorhen (*Gallinula chloropus*; n = 1 nests), marsh harrier (*Circus aeruginosus*; n = 1 nest), little bittern (*Ixobrychus minutus*; n = 1 nest), purple heron (*Ardea purpurea*; n = 1 nest) and Savi´s warbler (*Locustella luscinioides*; n = 3 nests). Although the dry-nesting species varied in potential contact with water (e.g., duck species are in frequent contact with water, while the marsh harrier and Savi's warbler have almost no water contact), we could not further classify nesting strategies due to low sample sizes in dry-nesting birds. In total, we sampled 101 egg and 24 water samples (total 125 samples). All samples successfully produced cell cultures (i.e., total sample size of 125 samples). In contrast, the success of our DNA extraction for the genetic characterisation of microbiota was more variable, resulting in reduced sample sizes for these analyses. This corresponded to 73 egg and 22 water samples for all genetic analyses. See S2 Table for further details on sample sizes.

### Bacterial sampling procedure

We used two techniques to quantify eggshell-associated bacteria in our study – we firstly cultured bacteria on selective and non-selective plates, and then genetically-assessed bacterial assemblages using ARISA [automated ribosomal intergenic spacer analysis; 19]. A principal advantage of bacterial culture techniques is that they provide information on the abundance of specific bacterial taxa and are therefore useful for quantifying differences in bacterial loads between treatment groups [e.g., 12]. However, only a small proportion of bacteria can be grown in laboratories [20] and cultured-based techniques are therefore limited in their ability to quantify the diversity of bacterial assemblages. In contrast, molecular techniques, while potentially less-reliable in estimating bacterial loads [for example, due to biases in PCR amplification: 21], can provide a more comprehensive representation of bacterial diversity. For example, ARISA detects bacterial

diversity by amplifying and sizing the intergenic spacer region (IGS) between 16S and 23S rRNA genes, which can estimate the number of operational taxonomic units (OTUs) in a microbial sample [see below; 22]. Nonetheless, ARISA can still only provide an estimate of diversity due to issues associated with taxon-specific differences in genomic structure (e.g., some species do not have the 16S and 23S genes organised in an operon), leading to some bacteria being undetected and others over-represented [e.g., 22]. Combining culture-based and molecular techniques can therefore provide a more comprehensive overview of bacterial loads and diversity than using each technique separately. Despite recent advances in the molecular characterisation of bacterial communities, which allow finer resolution of bacterial assemblages [e.g., metagenomic techniques; 23], ARISA and culture-based techniques remain effective tools to characterise broad differences in assemblages between treatment groups [e.g., 24,25].

Bacterial samples were taken from egg shells and water samples. To test whether bacterial assemblages on wet-nester eggs were similar to assemblages found in the surrounding water, we collected 24 water samples from the four study sites. Water was stored in sterile 10 ml tubes, of which 100μl was used for bacterial cultivation.

To sample eggshell bacteria, we removed one randomly-selected egg from the nest with sterile gloves on the day the nest was found and swabbed the whole egg surface for 17 seconds with a sterile cotton swab, soaked in sterile water. The swab was stored in a transport medium (transport viscose swab with Amies transport medium, Sarstedt) and, within 48 hours, was transferred to a 15 ml tube containing 2.5 ml of PBS buffer. The sample was mixed by vortex agitating for 120 seconds.

A portion of the bacterial suspension was used for genetic analysis (see below), while 100 μl of each bacterial suspension was subjected to serial decimal dilutions. Dilutions ranged from no dilution to $10^{-6}$, and 0.1 ml of each diluted sample was then plated onto separate agar plates (see below). Typically, four different concentrations were prepared for each sample. After 24 hours of incubation at 37°C, bacterial colonies emerged and grew, each originating from a single founder bacterium. Consequently, each colony-forming unit (CFU) represented an individual bacterium. The plate with the optimal number of colonies for distinct separation (usually up to 300 colonies per plate, based on our experience) was selected. CFU abundance was counted directly from plates when CFU density was low. We took a photograph of high-density plates, magnified the image on a screen and marked each CFU as it was counted. The total CFU count for all plates was then extrapolated based on the dilution factor of the corresponding plate.

Bacterial samples were spread-plated on the 1) non-selective media (Columbia agar with sheep blood 7%, company Oxoid), which allowed the distinction between haemolytic and non-haemolytic bacteria and 2) selective media (Brilliance UTI, company Oxoid), which allowed the detection of the principal micro-organisms causing urinary tract infections in humans (*Enterococcus* spp., *Escherichia coli*, *Proteus*, *Morganella*, *Providencia*, *Pseudomonas*, *Staphylococcus*, *Streptococcus*, coliforms). These taxa are also associated with a variety of infections in birds [e.g., 8,26,27].

## Genetic analyses

Bacterial DNA was extracted using a Qiagen DNeasy Blood & Tissue Kit, following the manufacture's protocol for the purification of total DNA from bacteria. Although this protocol isolates DNA from both Gram-positive and Gram-negative bacteria, a pretreatment for Gram-positive bacteria was included, as these bacteria have thicker cell walls than Gram-negative bacteria. We also included several control samples in our DNA extractions (i.e., replacing samples collected in the field, with sterilised water from the laboratory) to identify potential contaminants introduced into the samples during the DNA extraction process. To characterise bacterial assemblages present in each sample we used ARISA [19], a commonly-used technique in microbiological studies to characterise bacterial assemblages [e.g., 12,28–30]. ARISA involves PCR amplification of the bacterial 16S-23S IGS using a fluorescently-labelled primer, after which high-resolution electrophoresis occurs using an automated system. Assemblages are therefore characterised by a series of electrophoretic peaks that vary according to the length of the amplified IGS fragment of each bacterial OTU (operational taxonomic unit).

We performed ARISA following the protocols outlined in Brandl et al. [12]. Each ARISA profile was subsequently viewed in Genemapper 4.0 (Applied Biosystems), recording the fragment length and intensity of each electrophoretic peak. The DNA fragment amplified by our primers consisted of the bacterial IGS, as well as approximately 20 bp and 130 bp of the 16S and 23S rRNA genes, respectively. Electrophoretic peaks of less than 270 bp were considered PCR artefacts, which was a similar cut-off to previous studies [e.g., 1,31].

## Statistical analysis

**Cultured bacterial abundance.** To compare cultured bacterial loads between eggs from wet and dry nests, we first conducted a generalised linear mixed models (GLMMs) in IBM Statistics SPSS 29.0, assuming a gamma distribution with a logarithm link. We pooled data for all thirteen species and analyses were conducted on bacterial taxa identified using both the non-selective (haemolytic and non-haemolytic bacteria) and selective (UTI bacteria) plates. Due to the repeated nature of the sampling, we incorporated species and sampling location (i.e., the four different waterbodies) as random effects.

Due to the low sample sizes for most species, we did not control for additional potentially-confounding variables in our analyses to avoid overfitting our models [32]. However, factors such as sampling date, or number of eggs in the nest, may also affect our estimates of bacterial abundance. Firstly, as environmental variables such as ambient temperature and humidity vary seasonally, which in turn can influence bacterial growth on eggshells [e.g., 33], sampling date may influence bacterial loads. Secondly, the number of eggs in a nest may affect horizontal transmission of bacteria, with more points of contact and potentially higher bacterial transmission of bacteria in nests with more eggs. In addition, egg incubation is also known to influence egg shell bacteria [12], but only commences after a certain species-specific number of eggs are laid. Although we had no data on incubation in the nests sampled [the onset of incubation can be difficult to determine based on clutch size due to partial incubation; 34], incubation is more likely in nests with a greater number of eggs. To test whether sampling date or number of eggs influences our estimates of bacterial abundance, we conducted additional GLMMs for the two species for which we had sufficient sample sizes –the common coot (N = 28; a dry-nester) and great-crested grebe (N = 43; a wet-nester). Sampling day was calculated as the number of days from April 1. Sampling location was included as a random effect, species as a fixed effect and sampling date and number of eggs as covariables. We again assumed a gamma distribution with a logarithm link.

**OTU assemblage structuring.** The structure of bacterial assemblages was characterised using the community ecology software PRIMER v6.1.6 [35]. Before the analyses were conducted we controlled for intersample differences in DNA extraction efficiency and PCR amplification success by dividing the intensity of each peak within an ARISA profile by the sum of all peak intensities within that profile [36]. Assemblage diversity was calculated for each egg (wet and dry) and water sample using the total number of OTUs detected, and the Shannon diversity index ($H' = -\sum \rho_i \ln \rho_i$, where $\rho$ represents the proportion of the total abundance arising from the $i$th OTU).

We calculated a zero-adjusted resemblance matrix for all samples using the Bray-Curtis similarity measure. This technique calculates compositional differences between assemblages independent of species relatedness, but does not control for phylogeny amongst the thirteen study species. Although bird phylogeny may have an influence on bacterial assemblages, our results suggest minimal clustering of bacterial assemblages occurs with species identity (see nMDS analyses in Results) and phylogenetic signals in similar studies have been very low [e.g., 15].

Clustering of bacterial assemblages by study site, nest-type (i.e., wet or dry nest) or species was visualised using a non-metric multi-dimensional scaling analysis (nMDS). The fit of the data was assessed via the stress values associated with the nMDS, with a stress of less than 0.2 being acceptable. Data were mapped into two- or three-dimensional space depending on whether the stress of the two-dimensional nMDS was lower than 0.2.

While nMDS provides a visual representation of clustering, it does not statistically test whether similarities between samples in a predetermined group are greater than expected by chance. We therefore implemented PERM to statistically

compare similarities in bacterial assemblages within our predetermined groups (i.e., study site, nest-type or species) with similarities in bacterial assemblages within groups, where samples were randomly allocated. PERM uses matrices of a pairwise relatedness statistic ("Sxy" – which corresponds to Bray-Curtis similarity in this study) and calculates the sum of all Sxy values (i.e., Bray-Curtis sums) for each group. This value is then compared to a distribution of Sxy sums generated from randomly assigning samples to groups. 1,000 permutations were used.

We tested several hypotheses with the nMDS and PERM analyses. Firstly, we were interested in whether bacterial assemblages clustered by sampling location, using groups for which we had relatively high sample sizes (i.e., water, great-crested grebe and common coot). We then tested whether samples clustered depending on sample type (i.e., pond water vs eggs), nest-type (i.e., eggs from wet vs dry nests) and species. Finally, we were interested in whether the bacteria found on eggs from wet nests are influenced to a greater extent by bacteria present in pond water than were eggs from dry nests. We therefore used the Bray-Curtis distance between each egg sample and the most closely-matched water sample by location and date (N = 62 samples). We then conducted a GLMM to test whether the mean Bray-Curtis distance between these egg-water dyads was shorter for wet nest eggs than for dry nest eggs. Bird species was used as a random effect and data followed a gamma distribution with a logarithm link. All data are presented as means ± SE.

**Statistical analysis of water samples.** It was not possible to sample bacteria from water and egg samples using the identical method (i.e., eggs were swabbed for 17 seconds, while 100µl of water sample was used for water-borne bacterial assemblages). Caution is therefore required in interpreting analyses incorporating water-borne bacteria. For example, although the diversity of bacteria sampled may not be affected by sampling technique, it is likely that bacterial abundance depends on which technique is used. As nMDS is based more on OTU identity than abundance, we included water samples in these analyses. However, the bacterial cultures are based entirely on abundance, and we therefore did not include water in these analyses.

## Results

### Cultured bacterial abundance

Across all samples, haemolytic bacteria comprised 18.3% of bacteria grown on the non-selective plates. On the UTI plates, *Enterococcus* (26.6%) was the most prevalent, followed by coliforms (24.4%) and *Staphylococcus/Streptococcus* (13.8%). The remaining CFUs (35.2%) could not be reliably identified by colour. Bacterial loads varied both within and across species (S3 Table). Compared to dry-nesting species, wet-nesting species exhibited a higher abundance of eggshell-associated bacteria across the majority of bacterial taxa (Table 1 and Fig 1). Similarly, when comparing the

**Table 1. Generalised linear mixed model output on the effect of nest-type (wet and dry) on the abundance of colony forming units found on eggshells in the thirteen study species. Separate GLMMs were conducted for each bacterial taxon. Coefficients for wet nests were set to zero.**

| Intercept | | | Nest-type | | |
|---|---|---|---|---|---|
| Coefficient | t | p | Coefficient (dry/wet) | t | p |
| Haemolytic bacteria | | | | | |
| 15.035 ± 1.683 | 8.932 | <0.001 | −7.419 ± 2.025/0 | −3.664 | <0.001 |
| Non-haemolytic bacteria | | | | | |
| 16.777 ± 0.888 | 18.887 | <0.001 | −5.530 ± 1.039/0 | −5.324 | <0.001 |
| *Enterococcus* sp. | | | | | |
| 14.985 ± 1.530 | 9.793 | <0.001 | −4.012 ± 1.738/0 | −2.308 | 0.023 |
| Coliforms | | | | | |
| 13.506 ± 3.882 | 3.479 | <0.001 | −0.832 ± 4.767/0 | −0.175 | 0.862 |
| *Staphylococcus/Streptococcus* sp. | | | | | |
| 14.325 ± 1.221 | 11.819 | <0.001 | −3.351 ± 1.413/0 | −2.372 | 0.020 |

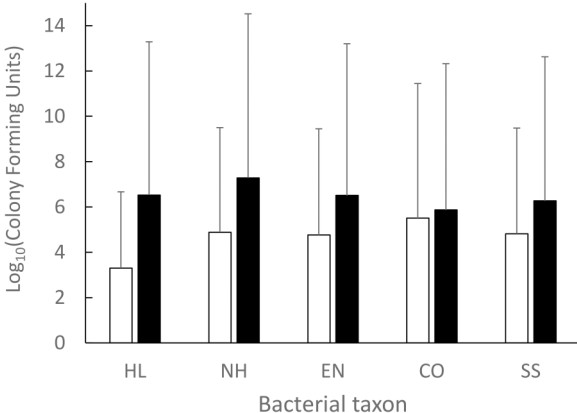

**Fig 1. Mean logarithmic abundance (±SE) of colony-forming units grown from eggshell bacterial swabs from the thirteen study species.** Bacterial taxa are haemolytic bacteria (HL), non-haemolytic bacteria (NH), *Enterococcus* species (EN), coliforms (CO) and *Staphylococcus*/*Streptococcus* species (SS). Dry-nesting species are represented by white, and wet-nesting species are represented by black bars. Asterisks indicate statistical significance, including non-significance (ns), p<0.05 (*) and p<0.001 (***).

common coot (dry nester) with the great-crested grebe (wet-nester), the two species for which we had the most robust sample sizes, bacterial abundance was highest for the wet-nester, even after controlling for number of eggs in the nest and sampling date (S4 Table). Bacterial abundance did not vary with the number of eggs in the nest, but did increase with sampling date for coliforms only.

## OTU diversity in wet and dry nests

We identified seven OTUs in the control samples, which were assumed to be contaminants. Each contaminant was present in 15.0±33.4% of samples (range=1.0–90.7% of samples) and all ARISA peaks corresponding to potential contaminants were not considered in subsequent analyses. In total, we detected 260 OTUs in all samples, which ranged in size from 271 to 1222 bp. No OTUs were detected on the eggs of all species (mean percentage of species in which each OTU was detected=29.7±1.2 SE %; range=9.1–90.9%), while 66 OTUs (25.3%) were detected in only single species. In addition, 37 OTUs (14.2%) were only found in water samples. Within species, some OTUs were abundant and present in all samples, while the majority of OTUs were detected in a small number of samples (S5 Fig).

There was no difference in the number or diversity of OTUs detected on wet eggs, dry eggs or in the water samples (number of OTUs: dry eggs=20.9±2.2 OTUs, wet eggs=22.7±4.0 OTUs, water=27.3±3.3 OTUs, $F_{2,92}$=1.279, P=0.283; Shannon diversity: dry eggs=2.2±0.1, wet eggs=2.3±0.2, water=2.6±0.2, $F_{2,92}$=2.114, P=0.123).

## OTU assemblage structuring

The nMDS analysis revealed no significant clustering of bacterial assemblages by site for water samples (water: observed similarity sum: 5574; expected similarity sum: 5555±77 SD; N=21, p=0.395; S6a Fig). However, spatial variation in bacterial assemblages was detected for both the common coot (observed similarity sum: 10124; expected similarity sum: 9774±105 SD; N=25; p=0.003; S6b Fig) and the great crested grebe (observed similarity sum: 4334; expected similarity sum: 4167±48 SD; N=21; p=0.003; S6c Fig).

In contrast, we detected substantial differentiation in the bacterial assemblages between sample types (Fig 2). First, bacterial assemblages in water samples were markedly different from assemblages on eggs and bacterial assemblage similarity was greater within sample types (i.e., water or egg samples) than between sample types (observed similarity sum: 161,772; expected similarity sum: 157,435±339; N=95; p<0.001). Second, egg bacterial assemblages clustered

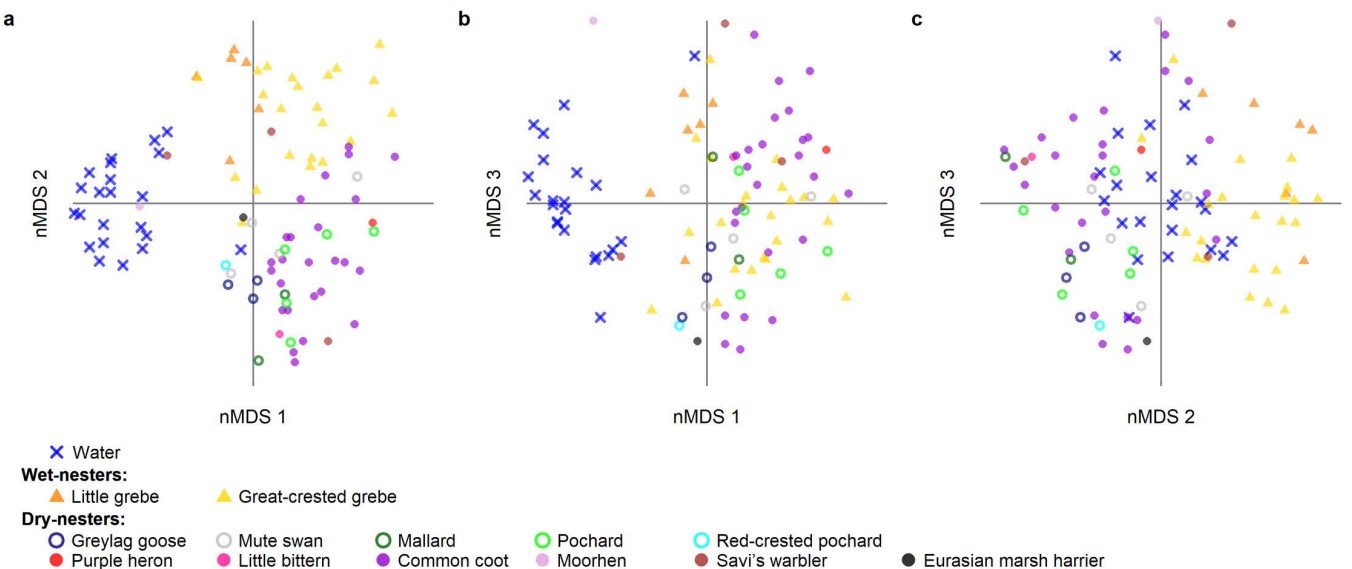

**Fig 2. Non-metric multi-dimensional scaling along a) the first and second axes, b) the first and third axes and c) the second and third axes, outlining differences in bacterial assemblages between the eggs of various waterbird species and water samples (nMDS stress = 0.19).** The symbols in the figure legend indicate species identity and nesting strategy (circles for dry-nesters and triangles for wet-nesters), while the water samples are represented by cross symbols. Species of the family Anatidae (ducks, swans and geese) are highlighted using open circles.

by bird species across wet- and dry-nesters (observed similarity sum: 37113; expected similarity sum: 35435 ± 195 SD; N = 73; p < 0.001), although clustering by species only occurred for the two wet-nesting species (Fig 2). Lastly, assemblages clustered depending on whether they occurred in wet or dry nests. Accordingly, eggs were more similar to other eggs within the same nest-type (i.e., wet or dry nests) than expected by chance, regardless of bird species identity (observed similarity sum: 88160; expected similarity sum: 85280 ± 233 SD; N = 73; p < 0.001). However, eggs in wet nests did not have more similar bacterial assemblages to the surrounding water than eggs in dry nests (GLMM: $F_{1,60}$ = 0.070, P = 0.792).

## Discussion

We found possible non-random clustering of bacterial assemblages on eggshells in wetland-nesting bird species. While we found only limited evidence for clustering of bacterial assemblages on eggshells by sampling locations, the more pronounced clustering appeared to occur between species with different nesting strategies. Specifically, eggshell bacterial assemblages sampled from the two closely related wet-nesting species (i.e., the great-crested grebe and little grebe) were markedly different from bacteria on eggshells from our broad category of dry-nesting species. Finally, bacterial loads were generally much greater on wet-nesting eggs, even when limiting analyses to species for which we had a robust sample size (where we controlled for clutch size and sampling date). Our data do not allow us to distinguish the contributions of horizonal and vertical transmission on the assemblage characteristics of eggshell-associated bacteria in our study. However, the association between nesting strategy and eggshell bacterial communities suggest that environmental sources may be a significant contributor of eggshell bacteria in our study species, although further research, incorporating more comprehensive sampling, is required to confirm this. Overall, these results suggest that nesting strategy may be an important determinant of bacterial assemblages and that conditions on egg surfaces may select for different bacterial assemblages.

The aim of our study was to test whether bacterial abundance and assemblage structure vary based on nesting-strategy. The sampling unit was therefore 'species identity' rather than 'number of individuals per species' which is instead treated as a random effect in our analyses (see below). Our sample sizes are higher for our broad category of dry-nesters (eleven species), while we only include two closely-related wet-nesting species in our analysis, due to the limited number of wet-nesting species in the study area. In addition, samples sizes within species were often limited. This restricts the range of analyses that we can perform on our data, and could mask true differences in bacterial assemblages with nesting strategy. Despite this, our findings still suggest novel differences in both bacterial abundance and assemblage structure between wet- and dry-nesters, at least in the thirteen species studied here. Future studies would benefit from including more wet-nesting species and more individuals per species. This will allow a more detailed and robust exploration of the determinants of fine-scale variation in bacterial abundance, such as types of nesting materials used, incubation behaviour and temporal influences on microbiota. Additionally, sampling more than one egg per clutch would be beneficial to account for intra-clutch variation in eggshell microbiota (e.g., due to egg laying order, partial incubation or brood parasitism), while using candling techniques to estimate embryo age within each egg [e.g., 37] would help account for microbial variation due to incubation stage. Interestingly, our analyses on the two species for which we have robust sample sizes (common coot and great-crested grebe) demonstrate that, even when controlling for temporal effects and clutch size, differences in bacterial loads are still detected between the wet-nesting (great-crested grebe) and dry-nesting (common coot) species.

The eggs of the two wet-nesting species in our study are typically in contact with the water within the nest or in soaked nesting material. The parents also cover the eggs with wet plant material when they leave the nest [18]. Wetland water harbours a high diversity and abundance of bacterial species, many of which are crucial for nutrient cycling [38]. This suggests the possibility of horizontal transmission of bacteria between the non-sterile feathers of adults and nesting material to the incubated eggs. In addition, bacterial growth increases with moisture [9,16]. Together, the moist nest conditions and possibility of horizontal transmission could explain the higher loads across most bacterial taxa of wet-nesters compared to dry-nesters, although it remains unclear why coliform bacteria did not differ between wet- and dry-nesters. Our findings are consistent with the findings of Darolová et al. [17] who observed higher bacterial loads on eggshells in wetland-species compared to terrestrial species. Our study suggests that variation also exists *within* wetland-associated species and that, in these species, the degree of contact of eggs with wetland water influences eggshell microbiomes.

Interestingly, the bacterial assemblages of our broad category of dry-nesting species clustered together, despite the marked differences in their behaviour, nest architecture and ecology. For example, ducks, such as the mallard and red-crested pochard, swim and forage in the water and therefore have intensive water contact. In addition, ducks are known to line nests with down-feathers [e.g., 39], which may further influence eggshell bacteria. In contrast, the little bittern and purple heron mainly have water contact with their legs, while the marsh harrier and Savi´s warbler have almost no contact with water. The architecture of nests and placement within the breeding sites also differ markedly between these species. Finally, some species may display intermittent incubation [e.g., 34] or experience brood parasitism [40], which can further complicate interspecies explorations of eggshell-associated bacteria. Despite these differences, our clustering analyses suggested similarities in bacterial assemblages within dry-nesters. In addition, our nMDS analyses suggested no obvious clustering of bacterial assemblages of the family Anatidae, which includes ducks, geese and swans, despite their unique nesting behaviours (e.g., intermittent incubation). These patterns support the possibility that nest microhabitat has a stronger influence on egg bacterial composition than bird phylogeny or ecology. However, larger sample sizes for each species are necessary to validate this finding and to further explore whether nest architecture within dry-nesters also influences bacterial assemblages on eggshells. The two wet-nesting species both belong to the family Podicipedidae, and it is therefore more difficult to disentangle the effect of shared phylogeny from shared microhabitat in influencing eggshell microbial communities. Future studies would benefit from more thorough sampling of dry-nesting species, plus including additional wet-nesting species, as only two such species nested at our study sites.

Our nMDS analysis suggested very little overlap in the bacterial assemblages between the water samples and eggshells of the wet-nesting species. While this pattern may reflect ecological filtering (see below), the observed dissimilarity may also be due to the field sampling and genetic techniques adopted for this study. Firstly, bacterial sampling from eggs involved eggshell swabbing, while water-associated bacteria were quantified via a 100µl water sample. This difference in sampling technique may have influenced our estimates of bacterial abundance (but less likely our estimates of number of distinct OTUs). Secondly, the ARISA method used to genetically characterise bacterial assemblages carries inherent biases, such as limited taxonomic resolution [e.g., 22], which may limit our ability to identify all OTUs present in our samples. Therefore, as the Bray-Curtis dissimilarity index used in the nMDS analysis incorporates both OTU identity and abundance, sampling issues may obscure true variation in bacterial assemblages associated with sample type.

Alternatively, the observed dissimilarity between eggshell-associated and water-associated bacteria may be due to ecologically-significant mechanisms that influence which water-borne bacterial assemblages are able to grow on the eggs of wet-nesting species. For example, egg incubation is known to influence eggshell bacterial assemblages by decreasing humidity and is therefore an important behavioural mechanism to reduce trans-shell infection of pathogens [12,41,42]. Likewise, the antimicrobial properties of nesting material [10,43] and the nanostructure of eggshells [13] are also known to influence the structure of associated bacteria assemblages. Finally, eggs contain antimicrobial substances, such as lysozyme and ovotransferrins which are known to display anti-microbial properties [11], although Darolová et al. [17] found no difference in maternal egg immune defence between wetland-associated and terrestrial bird species. These adaptations are important, as trans-shell penetration of eggshell bacteria can occur, which may increase embryonic mortality. For example, experimental inoculation of eggs with eggshell-associated species has been shown to increase embryonic mortality [7], although other studies have found no consistent link between trans-shell bacterial infection and hatching success [e.g., 44]. Although these mechanisms may play a role in shaping eggshell-associated bacterial communities, additional targeted research is needed to demonstrate a link between nesting strategy and antimicrobial adaptations on eggs in our study species.

Overall, our data suggest that nesting strategies of various bird species within a single habitat may influence eggshell bacterial assemblages. An increase in abundance of eggshell bacteria of several magnitudes was detected in the two closely-related wet-nesting species, compared to our broad category of dry-nesting species. Definitive conclusions regarding the relationship between nesting strategy and egg-associated bacterial assemblages will, however, require further research incorporating more robust field sampling and genetic analyses. Future studies would, for example, benefit from increased sample sizes, both within and across species, to allow a robust exploration of not only nesting strategy on eggshell microbiota, but other ecological factors such as egg laying date, incubation, and nesting materials. Moreover, future research should adopt more advanced molecular approaches, such as 16S rRNA gene sequencing and metagenomics [e.g., 23,45], to enable a deeper and more accurate characterisation of microbial communities.

## Supporting information

**S1 Fig.  Wet and dry nests located at the study sites.** Species are a) great-crested grebe (wet-nester), b) common coot (dry-nester) and c) mute swan (dry-nester).
(DOCX)

**S2 Table.  Sample sizes for each study species and environmental water.** Samples sizes include number of nests sampled, samples used for bacterial culture and samples used for ARISA analysis.
(DOCX)

**S3 Table.  Descriptive statistics of cultured bacteria across thirteen study species and water samples.** Statistics include sample size (N), mean and standard deviation (X̄±SD) and Range (minimum – maximum). Sample types are wet-nester (WN), dry-nester (DN) and water sample (WS).
(DOCX)

**S4 Table. Generalised linear mixed model output on the effect of species (common coot [dry-nester] and great-crested grebe [wet-nester]), number of eggs in nest and sampling data on the abundance of colony forming units found on eggshells.** Separate GLMMs were conducted for each bacterial taxon. Coefficients for the great-crested grebe were set to zero.
(DOCX)

**S5 Fig. Prevalence of OTUs found on eggs of different bird species and in water samples.** Species are a) common coot (n = 27), b) moorhen (n = 1), c) pochard (n = 5), d) Eurasian marsh harrier (n = 1), e) great-crested grebe (n = 29), f) greylag goose (n = 3), g) little bittern (n = 1), h) little grebe (n = 4), i) mallard (n = 2), j) mute swan (n = 4), k) purple heron (n = 1), l) red-crested pochard (n = 1), m) Savi's warbler (n = 3), n) water (n = 22). Prevalence was calculated as the percent of samples in which each OTU was detected. Each vertical bar represents an OTU and the position of each OTU is identical within each graph.
(DOCX)

**S6 Fig. Non-metric multi-dimensional scaling along the first and second axes outlining differences in bacterial assemblages between sampling groups.** Sampling groups are a) water samples, b) common coot egg samples and c) great-crested grebe collected at different locations including Budkovany (black circles), Jakubov (blue circles), Velke Blahovo (white circles) and Vrable (green circles).
(DOCX)

**S7 Datafile. The data used to generate graphs displayed in this manuscript.** This file contains data used to generate Figs 1, 2 and S5 Fig.
(XLSX)

## Acknowledgments

We thank Lucia Rubáčová for assistance with field work and four anonymous referees for taking the time to provide helpful comments on this manuscript.

## Author contributions

**Conceptualization:** Wouter F.D. van Dongen, Hanja B. Brandl, Alžbeta Darolová, Herbert Hoi.

**Data curation:** Hanja B. Brandl, Alžbeta Darolová, Ján Krištofík.

**Formal analysis:** Wouter F.D. van Dongen, Herbert Hoi.

**Funding acquisition:** Alžbeta Darolová.

**Investigation:** Hanja B. Brandl, Alžbeta Darolová, Ján Krištofík.

**Methodology:** Alžbeta Darolová, Ján Krištofík.

**Writing – original draft:** Wouter F.D. van Dongen.

**Writing – review & editing:** Hanja B. Brandl, Alžbeta Darolová, Herbert Hoi.

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
