## [Decision Letter · Decision Letter 0]

27 Jan 2025

Dear Dr. van Dongen,

Thank you for submitting your manuscript to PLOS ONE. After careful consideration, we feel that it has merit but does not fully meet PLOS ONE’s publication criteria as it currently stands. Therefore, we invite you to submit a revised version of the manuscript that addresses the points raised during the review process.

We look forward to receiving your revised manuscript.

Kind regards,

Petr Heneberg

Academic Editor

PLOS ONE

Journal Requirements:

“This work was supported by the Scientific Grant Agency of the Ministry of Education, Science, Research and Sport of the Slovak Republic and the Slovak Academy of Sciences, VEGA (project number 2/0077/11).”

Reviewers' comments:

Reviewer's Responses to Questions

**Comments to the Author**

1. Is the manuscript technically sound, and do the data support the conclusions?

Reviewer #1: Partly

Reviewer #2: Partly

Reviewer #3: No

Reviewer #4: Partly

2. Has the statistical analysis been performed appropriately and rigorously?

Reviewer #1: No

Reviewer #2: Yes

Reviewer #3: No

Reviewer #4: N/A

3. Have the authors made all data underlying the findings in their manuscript fully available?

Reviewer #1: Yes

Reviewer #2: Yes

Reviewer #3: Yes

Reviewer #4: No

4. Is the manuscript presented in an intelligible fashion and written in standard English?

Reviewer #1: Yes

Reviewer #2: Yes

Reviewer #3: Yes

Reviewer #4: No

Reviewer #1: This study provides valuable evidence on how nesting strategies—wet-nesting and dry-nesting—affect both the quantity and community structure of bacteria adhering to the eggshells of wetland-dwelling bird species. By presenting empirical data that align with intuitive expectations, the study effectively demonstrates the impact of nesting strategies on eggshell bacterial communities, thereby contributing new insights to the field of avian microbial ecology. The progression from objectives and experimental methods to results and conclusions is exceptionally orderly and logically structured. This clarity enhances the significance of the research findings, emphasizing that the results are not merely speculative but hold meaningful evidence. However, there are several significant shortcomings.

Comments

1.It is regrettable that this study had a small sample size, ultimately failing to present statistically significant data. Due to the limited number of samples, a detailed evaluation of other environmental factors influencing bacterial communities (such as temperature, humidity, types of nesting materials, seasonal fluctuations, and annual variations) was insufficient. In future studies, it is recommended to increase the sample size and expand the number of samples for dry-nesting species to more accurately assess the impact of nesting strategies. Enhancing data reliability by increasing the number of dry-nesting species samples is crucial.

2.The authors independently reported results from both culture-independent (direct DNA extraction and ARISA analysis) and culture-dependent (bacterial culturing) methods. However, by detailing the relationship and differences between the bacterial communities detected via ARISA analysis and the culturing results, the complementary roles of both methods can be clarified. In addition, it would be beneficial to discuss the extent to which the culture-independent method captures bacterial diversity compared to the culture-dependent method, and whether there are any biases towards specific bacterial communities.

3.The introduction explains vertical transmission of eggshell bacteria in birds (transmission from the mother within the oviduct before eggshell formation). However, this study does not clearly distinguish whether the eggshell bacteria are derived from maternal contact or from other environmental sources. A discussion including comparisons with the bacterial communities present on the exterior and interior of the mother bird is necessary. Please discuss comparative analyses between the mother's bacterial communities (such as those in the cloaca or skin microbiota) and those on the eggshell to identify maternally transmitted bacteria.

4.The terms "wet-nesting" and "dry-nesting" are inconsistently hyphenated throughout the manuscript. It is recommended to standardize the hyphenation for consistency.

Reviewer #2: Summary:

In this manuscript, van Dongen and Brandl et al. seek to characterize the differences in microbial communities between dry- and wet-nesting bird species in wetland environs. The authors collected sample swabs from nests of both dry- and wet-nesting species to analyze the bacterial populations associated. They demonstrate an increased bacterial abundance on wet-nesting species eggs than dry-nesting eggs. They also use ARISA to show distinct bacterial populations among wet and dry-nesting eggs. Overall, this small study is meticulously controlled and reported, however lacks the depth which the field has come to expect from studies on microbial communities, and the exact techniques chosen are perhaps not ideal for the questions being asked. The authors should attempt to use more modern and accurate techniques to better dissect their hypothesis.

Major Comments:

• The authors determine bacterial abundance by selective and non-selective plating on bacterial growth plates. While this is fine for certain bacterial taxa, there is a risk of the isolated colonies not being representative of the actual microbial community, only the microbes capable of being cultured on agar. Performing 16S qPCR would be more accurately indicative of true bacterial abundance.

• The authors use automated ribosomal intergenic space analysis (ARISA) to detect and report OTUs from their sampled nests. While an established assay, ARISA sports few benefits over newer, more powerful techniques such as 16S sequencing or shotgun metagenomics. These techniques would allow the authors to better characterize the exact differences between the microbial communities under research.

Minor Comments:

• Line 144 – the word “media” is missing.

• Figure 1 – it would be helpful to include a legend in the figure itself indicating that white bars are dry-nesting birds and black bars are wet-nesting birds.

• Figure 1 – Statistical significance would be helpful to report in the figure itself using the * system for noting p-values.

• Figure 2 – Authors should note in the text the differences between figures 2a, 2b, and 2c.

Reviewer #3: Review: PONE-D-24-57432 - Bacterial assemblages on eggs reflect nesting strategies in wetland-associated birds

Dear Authors. When I read the abstract of your manuscript, I was excited about it because, as you mention, it is the first comparative study looking at the eggshell bacterial assemblage across wetland species with different nesting strategies – wet and dry. However, once I started reading the sampling methodology and statistical testing, my enthusiasm waned slightly. While I find the results regarding the differences in the bacterial community between water samples and on eggs of wet-nesting species mainly interesting supporting the strong nesting microhabitat effect of these wet-nesting strategy species, I have methodological doubts about the study that do not allow for the robust conclusions you state, e.g. that species identity does not play a large role in dry-nesting species. I think this result is more of an artifact of not having a study controlled for the age of the clutch from which you sampled eggshell microbiota from only one egg. Why not from the whole clutch, i.e. each egg, since you didn't control for clutch age and individual eggs at all? I consider this a major shortcoming and either you should add how you checked this, but since you do not mention the nesting stage at which you sampled the eggs at all in the "Material and Methods", rather you did not take this into account at all, you should discuss these limitations and poor sampling within the species sufficiently. I also don't consider it a correct approach to include species in the dry-nesting category that differ significantly in their nesting strategies - e.g. most species nest on the ground, while only a few have elevated nests. Also, ducks deposit their own down-feathers in the nest and also partially incubate the brood, whereas this is no longer typical of other species that represent this dry-nesting group. I suggest a couple of ways to reduce this variability in your data, at least for bacterial load/abundance testing, since you take this into account for ARISA-based OTUs community structure, but not for eggshell bacterial load testing.

Your discussion also needs thorough re-writing– mostly toning down your strong conclusions that are unfortunately based on inappropriate sampling and discussion of your sampling and methodological limitations.

I list my concerns and a possible partial solution below. I hopefully they will help to make your data and outcomes more robust.

Abstract:

Last sentence of abstract sounds a little vague. I understand what you want to express, but better will be use some sentence which stimulate the future research on antimicrobial protection of eggs across habitat humidity gradient or try to be more generalist in your expressing and make some future suggestions.

Material and Methods:

Line 128 – why you did not swab all egg in the nest to have some “common” bacterial assemblage?? Did you candle eggs to assign development stage of each nest-eggs? Difference between first and last egg in the nest may be 7 or more days – especially in the case of ducks. So if you take lastly laid egg (randomly) it can have lower bacterial load compared to first laid egg in the clutch. So, sampling eggshell of only one egg from clutch is really not good sampling approach. Please note, eggshell bacterial assemblage may significantly differ with respect to nesting/incubation period - bacterial assemblages on eggs reflect and vary in fresh or continuously incubated eggs. Moreover, considering that your sampling design is strongly imbalanced (e.g. 48 Great-crested grebe nests to compared to one Moorhen nest), I see the fatal flaw in your study as you did not control nest/egg age or incubation phase during the sampling period which was quite extensive (May-July). All these factors may strongly affect both, microbial load and assemblage. How you were able to control for this confounding effect in your study. Please justify. Otherwise, your results cannot be considered as reliable with omitting this variability into account and indices of bacterial load and communities may be very biased preventing comparison between individual nests (wet-, dry- breeders).

Line 144 – cultivation of bacteria – you did not run in replicates? How did you count CFU? Manually? Using SW from photography? Be more detailed in description of your technique?

- ARISA- please as this method is not such precise as sequencing, use term ARISA based-OTUs throughout the whole manuscript as term OTUs per se is emerged from more precise sequencing technique than ARISA.

Statistical analyses

To have you sampling design, I first inspect within species variability in bacterial load/abundnace. I am lacking basic exploratory data analysis (min, max, range, mean SD, SE – i.e. expressing variability of data within species. I guess it will be quite extensive. You can do at least for Fulica atra and Podiceps cristatus where you have nice sampling.

- another factor is covering the egg with the feathers – ducks (Mallards and Aythya spp.) use their own body feathers as nest lining material and cover eggs with these feathers. This might be other factor that can strongly affect eggshell microbial assemblage. To see your species and dataset. I am for to extend your statistics from glmer approach to also pairwise comparison of grebes or only Podiceps cirstatus with only Fulica atra as representative species from wet-, dry -nesting strategy continuum. I see grouping of ducks, Savi warblers and herons as not intuitive with respect to their nest building and nest material specifities. Also, most species are ground nesters but what about heron and Ixobrychus and Savi warbler??? I think you mixed too much together without appropriate controlling of confounding effects.

I am strongly for make more species groups – e.g. put ducks using feathers in the nest together as they really have specific nest building habits via incorporating own body down feathers into the nest and cover the eggs with them. You can do this statistic as additional testing that can uncover another specifities in eggshell bacterial load. Now I see you did for bacterial communities’ statistics, but why you did not do also for bacterial load (abundance) data??? E.g in Figure 1 I like to see boxplots where wet-nesters will be depictes separately, e.g. ducks also separately, and other species as well – except of these having n=1). I think it will be more reliable than to mix all dry-nesters together into one group consisting of many species with distinct nesting strategy.

Discussion

Line 316-320 – Sorry but this conclusion is too strong considering your sampling effort one to three nest per most of the dry-nesting species. Based on your dataset and fatal flaw with ignoring incubation phase in sampling egg, you cannot rule out the role of species identity on eggshell bacterial assemblage. Tone down this your conclusions and rather discuss thoroughly possible limitations of your study that more robust sampling controlled for nesting phase of sampling eggs in necessary to test these assumptions – i.e. rule out or test the effect of species identity on eggshell bacterial load. and discuss all these shortcomings of your study.

Lines 327-329 – also this conclusion needs to additional testing as I suggested. Use only dry-nesting species where appropriate number of individuals you have or cluster together ducks that have very specific nesting – incorporating of own body down feathers into the nest lining, egg covering and partial incubation..I think it will help to dilute all confounding effects that were omitted in your comparison and clustering all dry-nesting species in one group…

Lines 336 –337 - of course if you sample one to three nest per species and sample only one egg per clutch. It is not representative eggshell microbial assemblage per species, sorry, but you need more robust sampling to see some nesting ecology-based clustering.

Line 338 – yes, and also cover their eggs with nesting material full of down feathers that may have thermoregulatory and also antimicrobial properties. Also intermitently incubate their eggs before starting full incubating which affect both, eggshell, but also egg white bacterial load – see e.g. Effect of intermittent incubation and clutch covering on the probability of bacterial trans-shell infection | Semantic Scholar, or Laying date, incubation and egg breakage as determinants of bacterial load on bird eggshells: experimental evidence | Oecologia

You should take into account also this very specific nesting ecology before choosing species for your study…or at least take this into account during testing your hypotheses and discuss these specifications thoroughly.

Lines 343-344 and further – tone down thoroughly.. You really don´t have data on these strong conclusions.

Lines 346-348 – and also more thorough sampling of dry-nesting wetland species…

Lines 365-376 – but there is also studies that did not find and effect and in general there is no study testing direct effect of trans-shell penetration and embryo mortality. Discuss more precisely. It is not such straightforward as you described. Most of eggshell bacterial are not able to live inside the eggs etc. etc…

Line 357 – yes, incubation, and how did you control this in your study??? Try to put this infor into the context of your study. It is not discussion, just plain informative manner which you used here..

Line 366 – your findings may highlight something but are not such robust, unfortunately..So try to tone down that your findings may serve as some initial stimuli for further more thorough testing…

Reviewer #4: Specific comments:

-English proofreading is extremely recommended.

-You should double check the references style according the submission guidelines.

-Where are the references you followed for these experiments?

-What is the point of cultivating the samples on agar media that isn't selective?

-For clarity, please provide a concluding paragraph at the end of the study.

-In order to increase clarity and conciseness, I would like to combine the discussion and results sections into a single section. This is because the sections are extremely disorganized and lack sufficient data.

-Since you brought up the eggshells' antibacterial qualities (J L 44, 62, 72, 80, 360, and 362), your study never touched this point.

-In order to make the results so compelling and connected to the clinical findings, it would also be helpful to discuss the most common disease in these species or the chicks after hatching.

-Additionally, I would advise outlining how susceptible these bacterial species are to antibiotics in order to determine whether they are commensal or dangerous infections (MDR pathogens), against which certain bird species are immune.

-Why did you limit your attention to bacteria? How about fungi?

- Where is the raw date for the genetic findings?

-Fig 1: the error bars are so weired, please explain it.

-Please divide Table 1 into dry and wet categories with distinct data for each since it is worthless. Please explain what t/p stands for as well.

L48: Be specific and use birds instead of animals.

L82: Bacterial first, then genetic.

L180: Please provide an illustrative figure to demonstrate the steps taken in a sequential manner, as the sampling numbers are extremely unclear.

L195: Throughout the manuscript, use the same first line space.

L200: Findings? For clarification, please specify which figure.

L244: For wet or dry nesting samples, is 35.2%?

**Do you want your identity to be public for this peer review?** For information about this choice, including consent withdrawal, please see our Privacy Policy

Reviewer #1: No

Reviewer #2: No

Reviewer #3: No

Reviewer #4: No

---

## [Author Response · Author response to Decision Letter 1]

30 Mar 2025

Reviewer 1

Comment 1: In future studies, it is recommended to increase the sample size and expand the number of samples for dry-nesting species to more accurately assess the impact of nesting strategies. Enhancing data reliability by increasing the number of dry-nesting species samples is crucial.

Response: We agree with this comment, and comments made by other reviewers with respect to the relatively low sample sizes in our study. We acknowledge this limitation in both the Results and Discussion. In the Results, we state that our lower sample sizes limit the analyses we can complete (e.g. controlling for covariables such as sampling date, clutch size etc; lines 201-202). In the Discussion, we acknowledge that larger sample sizes are needed to validate our results (both an increase in wet-nesting species, and the number of individuals sampled per species. Lines 374 and 406-413). However, we also highlight that despite these lower sample sizes we are still able to demonstrate novel differences in bacterial abundance and diversity with nesting strategy in wetland associated birds, which is the first study to do so. Our findings provide valuable insights into the challenges associated with nesting in humid environments and it is hoped our research will stimulate further research into the consequences and adaptations of birds nesting in humid environments.

Comment 2: The authors independently reported results from both culture-independent (direct DNA extraction and ARISA analysis) and culture-dependent (bacterial culturing) methods. However, by detailing the relationship and differences between the bacterial communities detected via ARISA analysis and the culturing results, the complementary roles of both methods can be clarified. In addition, it would be beneficial to discuss the extent to which the culture-independent method captures bacterial diversity compared to the culture-dependent method, and whether there are any biases towards specific bacterial communities.

Response: We agree and now add a discussion in the Methods comparing the advantages and limitations of culture-based and ARISA techniques (lines 121-138). We argue that both techniques have their limitations, but together provide a more holistic overview of bacterial loads and diversity. We are not aware of any study that has shown that ARISA displays a bias towards specific bacterial taxa. We are happy to add such a reference should the reviewer be aware of one.

Comment 3: The introduction explains vertical transmission of eggshell bacteria in birds (transmission from the mother within the oviduct before eggshell formation). However, this study does not clearly distinguish whether the eggshell bacteria are derived from maternal contact or from other environmental sources. A discussion including comparisons with the bacterial communities present on the exterior and interior of the mother bird is necessary. Please discuss comparative analyses between the mother's bacterial communities (such as those in the cloaca or skin microbiota) and those on the eggshell to identify maternally transmitted bacteria.

Response: The mechanisms determining the acquisition of eggshell bacteria are of central importance to this study, although our data do not allow a distinction between environmentally- and maternally-acquired bacteria. We acknowledge that these two potential bacterial sources need to be mentioned, which we now do in the opening paragraph of the Discussion (lines 358-364). As our data do not permit a deeper exploration of horizontal vs vertical transmission (we did not swab maternal skin or cloaca), we do not provide further discussion on how different bacterial strains may be acquired via the two vectors.

Comment 4: The terms "wet-nesting" and "dry-nesting" are inconsistently hyphenated throughout the manuscript. It is recommended to standardize the hyphenation for consistency.

Response: Thank you. We have now gone through the manuscript and hyphenated all occurrences of the terms.

Reviewer 2

Comment 1: The authors determine bacterial abundance by selective and non-selective plating on bacterial growth plates. While this is fine for certain bacterial taxa, there is a risk of the isolated colonies not being representative of the actual microbial community, only the microbes capable of being cultured on agar. Performing 16S qPCR would be more accurately indicative of true bacterial abundance.

Response: We acknowledge the inherent limitations of culture-based methods and now discuss this in more detail in lines 121-138. We discuss the limitations of both culture-based techniques and ARISA, and that more advanced techniques exist. We also argue that, when used together, our methods still represent an informative tool in microbiology when focusing on broad differences in bacterial assemblages between treatment groups. Future studies would indeed benefit from using more modern techniques, such as 16S qPCR.

Comment 2: The authors use automated ribosomal intergenic space analysis (ARISA) to detect and report OTUs from their sampled nests. While an established assay, ARISA sports few benefits over newer, more powerful techniques such as 16S sequencing or shotgun metagenomics. These techniques would allow the authors to better characterize the exact differences between the microbial communities under research.

Response: We agree that the methods we have used here are now infrequently used due to advances in molecular techniques. We have acknowledged the inherent limitations of culture-based and ARISA techniques, but argue that despite these limitations, they remain an effective tool in characterising broad differences in bacterial communities between treatment groups. Indeed, we here have successfully used these techniques here to demonstrate that nesting strategy can influence eggshell bacterial communities in wetland-associated birds. We however recognize the potential benefit of integrating 16S qPCR in future studies to further validate our findings. We discuss these limitations in lines 121-138.

Comment 3: Line 144 – the word “media” is missing.

Response: ‘Media’ added

Comment 4: Figure 1 – it would be helpful to include a legend in the figure itself indicating that white bars are dry-nesting birds and black bars are wet-nesting birds.

Response: We prefer not to add a key, as this would add to cluttering within the figure. We have therefore left the explanation of the white and black bars in the caption (which will be located directly below the figure in the final publication).

Comment 5: Figure 1 – Statistical significance would be helpful to report in the figure itself using the * system for noting p-values.

Response: We have now added asterisks in the figure to denote significance levels.

Comment 6: Figure 2 – Authors should note in the text the differences between figures 2a, 2b, and 2c.

Response: We apologise, but we currently do not understand what the reviewer is requesting here. The three figures simply show the same MDS data along the three different axes. If the reviewer could clarify their suggestion, we would be happy to respond.

Reviewer 3

Comment 1: I have methodological doubts about the study that do not allow for the robust conclusions you state, e.g. that species identity does not play a large role in dry-nesting species. I think this result is more of an artifact of not having a study controlled for the age of the clutch from which you sampled eggshell microbiota from only one egg. Why not from the whole clutch, i.e. each egg, since you didn't control for clutch age and individual eggs at all? I consider this a major shortcoming and either you should add how you checked this, but since you do not mention the nesting stage at which you sampled the eggs at all in the "Material and Methods", rather you did not take this into account at all, you should discuss these limitations and poor sampling within the species sufficiently.

Response: After careful consideration, we agree with the reviewer that our analyses are not robust enough to determine if species identity play an important role in determining bacterial loads, given the low sample sizes for most species. We have therefore removed all previous discussion that nesting-strategy was a more important determinant of bacterial assemblage than species identity (throughout both the Abstract and Discussion). We also acknowledge the limitations associated with the low number of wet-nesting species in our study and the low number of individuals for which we have samples for dry-nesting species (lines 201, 374 and 406-413). Despite these limitations, we argue that we can still demonstrate the impact of nesting strategy on bacterial assemblages, which was the primary objective of this study.

We are unsure how our findings may be an artifact of not controlling for clutch age, as suggested by the reviewer. We are therefore happy to receive further feedback on this. However, we now show in the common coot and great-crested grebe, that the number of eggs in the nest and sampling date generally do not affect bacterial loads (except for in Coliforms, where bacterial abundance increases with sampling date; lines 213-219, 282-285 and S3 Table). This suggests that clutch age may not have a strong effect on eggshell bacterial communities, although further study is required to verify this.

With regards to our decision to only sample one egg per clutch, this was to maintain consistent methodology across species and to maximise the number of nests we could sample. The mean number of eggs per nest are highly variable between species, including great-crested grebe (3.8 eggs/nest), little grebe (5.4 eggs/nest), mute swan (5 .6 eggs/nest), common coot (7.4 eggs/nest) and mallard (9.2 eggs/nest). Sampling all eggs in all nests would result in significant variation in sample sizes between species and limit the number of nests we could sample due to resource constraints. We instead prioritised sampling a higher number of nests, rather than fewer nests but with more eggs. Finally, previous research has demonstrated consistency in egg-shell bacteria between, and hence presumably within, nests (e.g. Campos-Cerda, F., Torres, R., Nava, L. et al. 2023. Eggshell microbiome as a potential microbial reservoir in a cavity nesting bird. J Ornithol 164, 217–222). This latter result again suggests it would be more worthwhile sampling nests from a greater number of species, rather than more eggs within a single species.

Comment 2: I also don't consider it a correct approach to include species in the dry-nesting category that differ significantly in their nesting strategies - e.g. most species nest on the ground, while only a few have elevated nests. Also, ducks deposit their own down-feathers in the nest and also partially incubate the brood, whereas this is no longer typical of other species that represent this dry-nesting group. I suggest a couple of ways to reduce this variability in your data, at least for bacterial load/abundance testing, since you take this into account for ARISA-based OTUs community structure, but not for eggshell bacterial load testing.

Response: See our response below (Reviewer 2, Comment 10).

Comment 3: Your discussion also needs thorough re-writing– mostly toning down your strong conclusions that are unfortunately based on inappropriate sampling and discussion of your sampling and methodological limitations.

Response: As discussed throughout this reply to the reviewers, we agree that, based on our methodology, our conclusions were too strong in the original version of this manuscript. We have either toned down our conclusions, removed statements that are not supported by the data or added additional discussion to clarify our results. For example, we no longer include our earlier statements that species identity does not influence bacterial community (e.g. in the Abstract, and in the opening and concluding lines in Discussions) and acknowledge that larger sample sizes are required to validate our findings (lines 374 and 406-413).

Abstract:

Comment 4: Last sentence of abstract sounds a little vague. I understand what you want to express, but better will be use some sentence which stimulate the future research on antimicrobial protection of eggs across habitat humidity gradient or try to be more generalist in your expressing and make some future suggestions.

Response: We agree and now state: “Our research suggests that nesting in moist conditions presents unique challenges to wetland birds and highlights the potential for future research into the unique antimicrobial adaptations associated with the eggshells of these species” (lines 39-42).

Material and Methods:

Comment 5: Line 143 – why you did not swab all egg in the nest to have some “common” bacterial assemblage??

Response: Please see our response to Reviewer 3, Comment 1 (final paragraph).

Comment 6: Did you candle eggs to assign development stage of each nest-eggs? Difference between first and last egg in the nest may be 7 or more days – especially in the case of ducks. So if you take lastly laid egg (randomly) it can have lower bacterial load compared to first laid egg in the clutch. So, sampling eggshell of only one egg from clutch is really not good sampling approach. Please note, eggshell bacterial assemblage may significantly differ with respect to nesting/incubation period - bacterial assemblages on eggs reflect and vary in fresh or continuously incubated eggs. Moreover, considering that your sampling design is strongly imbalanced (e.g. 48 Great-crested grebe nests to compared to one Moorhen nest), I see the fatal flaw in your study as you did not control nest/egg age or incubation phase during the sampling period which was quite extensive (May-July). All these factors may strongly affect both, microbial load and assemblage. How you were able to control for this confounding effect in your study. Please justify. Otherwise, your results cannot be considered as reliable with omitting this variability into account and indices of bacterial load and communities may be very biased preventing comparison between individual nests (wet-, dry- breeders).

Response: We acknowledge that eggshell bacteria are likely to vary with egg age due to various factors, such as incubation. We also acknowledge that this temporal variation will likely add additional variability in eggshell assemblages that we have not quantified, although the exact day that incubation begins can be difficult to determine due to partial incubation (Wang and Beissinger 2011), which is now mentioned in lines 211-212. What we find fascinating is that we find clustering in bacterial assemblages between wet- and dry-nesters DESPITE these additional sources of variation. Indeed, due to our relatively low sample sizes (see discussion elsewhere in our response to reviewers), including additional covariables in our models (such as egg age) would likely lead to overfitting and potentially spurious results (see lines 201-219 for a discussion of this). Finally, our additional analyses on the great-crested grebe and the common coot, for which we had larger sample sizes, show minimal to no impact of the number of eggs (and thus the likelihood of incubation) or sampling date on bacterial communities (lines 282-285 and S3 Table). This further supports our decision to exclude additional covariables from our analyses. Future studies would benefit from collecting data from a larger number of nests and focusing on temporal variation in subsequent statistical analyses.

Jennifer M. Wang, Steven R. Beissinger, Partial Incubation in Birds: Its Occurrence, Function, and Quantification, The Auk, Volume 128, Issue 3, 1 July 2011, Pages 454–466,

Comment 7: Line 161 – cultivation of bacteria – you did not run in replicates? How did you count CFU? Manually? Using SW from photography? Be more detailed in description of your technique?

Response: We now state that “CFU abundance was counted directly from plates when CFU density was low. We took a photograph of high-density plates, magnified the image on a screen and marked each CFU as it was counted” (lines 157-159). We did not run replicates, but are confident that our estimates of abundance were accurate, due to the known repea

---

## [Decision Letter · Decision Letter 1]

7 May 2025

Dear Dr. van Dongen,

We look forward to receiving your revised manuscript.

Kind regards,

Petr Heneberg

Academic Editor

PLOS ONE

Journal Requirements:

Additional Editor Comments :

Academic Editor:

- While the introduction and discussion mention vertical transmission, the study does not provide data to distinguish it from environmental acquisition. This limits conclusions about maternal contribution to eggshell microbiota.

- Nesting strategy (wet vs. dry) is used as a broad grouping variable, but species within each group vary significantly in nesting materials, water contact, and incubation behavior. This undermines the strength of clustering conclusions based on nest type alone.

- One Egg per Nest Sampled**: Sampling only one egg per clutch does not account for intra-clutch variation, which can be substantial and could bias results - discuss this in the Limitations section.

- Bacterial communities change with incubation, and the study did not assess incubation stage or egg laying order. Without controlling for this, variability may mask true differences due to nesting strategy - discuss this in the Limitations section.

- Many dry-nesting species have sample sizes of 1–3 nests, making any comparison across species or nesting strategies tenuous - discuss this in the Limitations section.

- Use of Outdated Genetic Method (ARISA). While ARISA is a valid fingerprinting technique, it has been largely supplanted by 16S rRNA sequencing and shotgun metagenomics, which offer finer resolution. The authors acknowledge this but should clearly frame ARISA's limitations.

- Avoid Overstating Conclusions. Many conclusions regarding the effect of nesting strategy are too strong given the limitations in design. These should be reworded to reflect the exploratory nature of the study.

- Clarify Figures. Figures should include clearer legends and visual indicators (e.g., nesting strategy symbols or species groupings). Consider separating duck species from other dry-nesters in visualizations.

- Specify that future research should explore egg antimicrobial traits, control for incubation stage, and incorporate 16S sequencing to identify microbial taxa involved in pathogenic vs. commensal roles.

- Line 23: Change “Animals host diverse bacterial assemblages” to “Birds host diverse bacterial assemblages” to reflect study focus.

- Line 144: Add “the” before “media”: “…onto the media.”

- Hyphenation: Standardize the use of “wet-nesting” and “dry-nesting” throughout the manuscript.

- Tone Down Phrasing: Replace strong language such as “our findings are significant” with “our findings suggest” or “indicate.”

- Line 196: Improve clarity of sampling description—consider a flowchart or table to summarize sample numbers.

- Line 430-432: Phrase such as “highlight important differences” should be reworded to “suggest potential differences.”

- Ensure consistent italicization of all Latin species names throughout the manuscript, including the References.

- The manuscript references older papers for methods but does not cite newer alternatives (e.g., 16S sequencing), which should at least be discussed.

Reviewers' comments:

Reviewer's Responses to Questions

**Comments to the Author**

Reviewer #1: All comments have been addressed

Reviewer #3: (No Response)

Reviewer #4: All comments have been addressed

2. Is the manuscript technically sound, and do the data support the conclusions?

Reviewer #1: Yes

Reviewer #3: Yes

Reviewer #4: Yes

3. Has the statistical analysis been performed appropriately and rigorously?

Reviewer #1: Yes

Reviewer #3: Yes

Reviewer #4: Yes

4. Have the authors made all data underlying the findings in their manuscript fully available?

Reviewer #1: Yes

Reviewer #3: Yes

Reviewer #4: Yes

5. Is the manuscript presented in an intelligible fashion and written in standard English?

Reviewer #1: Yes

Reviewer #3: Yes

Reviewer #4: Yes

Reviewer #1: (No Response)

Reviewer #3: Dear authors. I am fully satisfied with the revised version of your manuscript. I think you improved the clarity and comprehensively toned down your conclusions and now the study is really useful as a stimuli for futures studies focused on eggshell bacterial assemblages and/or bacterial trans-shell penetration in the context of breeding ecology of birds.

I have only a few minor comments and suggestions which you can find below:

Technical comment:

Why you changed corresponding author in the revised version of the manuscript and why one of the co-authors (L. Rubáčová) has been removed from the author team to Acknowledgements only? Did she agree with this and it was discussed with her thoroughly? It is not usual such dramatically change author membership and status of authors?

To your comments regarding clutch age (my (Reviewer 3) Comment 1) and effect on eggshell bacterial assemblage. Yes, you considered sampling date and describe that effect might be linked with changed environmental factors. But, I am considering incubation of the clutch and resulting eggshell assemblage. As you described on lines 150-152, you sampled one egg from nest when you found it, but although you record the precise date of nest finding, you were not aware of the age of the clutch (i.e. how long the clutch has been incubated already). Moreover, it is quite common that 1) nesting parasitism occurred in Mallards and also coot and grebes and also that 2) intermittent incubation of partial not-completed clutch occurred before the full incubation and clutch completion (at least in ducks it is very common, and I think also swans or grebes are known for this behaviour). In practice, simple fieldwork “candler” (inspection of egg inner structure under the light) is used for determination of embryo age which correspond to intensity of incubation. And as you mentioned in you text of the revised ms that incubation may affect eggshell bacterial assemblage (and quite a lot), this discrepancy in eggs age and developmental stage within the clutch may cause bias if your chosen only one egg from the complete clutch for sampling. I think you should somehow touch of this possible bias in your data caused by the different clutch age, role of intermittent incubation and/or brood parasitism in your focal species.

But in general, it is nice you added additional GLMMs testing for the effect of clutch size and sampling date, as it is really highly probably that enlarged clutches may be incubated before clutch completion or being parasitized by brood parasites. This additional GLMMs made your conclusions more robust. Yet, still, as your study is pioneering in study of bacteria on eggshell with respect to breeding ecology, I am for to include possible factors that might be behind the differences between dry- and wet-nesters. Although I understand your point of view and you want to keep the line of the study as simple as possible, I thin a few sentences in the discussion with other potential factors cannot be distracting from the major message of your study.

Line 141 – “more holistic overview”., I suggest to replace holistic by “comprehensive”. Holistic is really to strong meaning in your case study.

Line 142 - …in isolation. I suggest to replace by “separately” or some other equivalent.

Line 290 – “….of bacteria on eggshells on wet-nesting..“ rephrase it please.

Line 295 – “coliforms” without capital please

Lines 359-366 – I am still wondering if it cannot be partially caused by pre-incubation behaviour of dry-nesters – covering of clutch by down feathers in mallard and other duck species, intermittent incubation I have mentioned above..I think you should take this into account in discussion of your results (mostly differences between dry and wet-nesters). You already did for covering the clutch by feathers, and even if you disagree to do for incubation (see response to my Comments 18). I think it can help to untangle the all possible confounding factors that have to be taken into account in future studies of eggshell or trans-shell bacterial assemblages.

Line 380 The aim of this study…..I suggest to rephrase… “…bacterial abundance and community structure vary based on……”.

Your Response to Comment 17 of Reviewer 3 – for example this study has show no effect of bacterial trans-shell penetration on hatching success: Effect of intermittent incubation and clutch covering on the probability of bacterial trans‐shell infection - Javŭrková - 2014 - Ibis - Wiley Online Library

Reviewer #4: After reviewing the revisions and responses, I find that all my comments have been adequately addressed, and I have no further comments.

**Do you want your identity to be public for this peer review?** For information about this choice, including consent withdrawal, please see our Privacy Policy

Reviewer #1: **Yes: ** Akihiko UDA

Reviewer #3: No

Reviewer #4: No

---

## [Author Response · Author response to Decision Letter 2]

24 Jul 2025

Academic Editor

Comment 1: While the introduction and discussion mention vertical transmission, the study does not provide data to distinguish it from environmental acquisition. This limits conclusions about maternal contribution to eggshell microbiota.

We agree that we cannot make firm conclusions on the maternal contribution to egg shell microbiota, nor tease apart vertical versus horizontal effects. Considering this caveat, we have modified the first paragraph in the discussion to better highlight this (lines 364-369), where we now state that “Our data do not allow us to robustly distinguish the contributions of horizonal and vertical transmission on the assemblage characteristics of eggshell-associated bacteria in our study. However, the association between nesting strategy and eggshell bacterial communities suggest that environmental sources may be a significant contributor of eggshell bacteria in our study species”. We hope that this statement sufficiently emphasises this limitation of our study.

Comment 2: Nesting strategy (wet vs. dry) is used as a broad grouping variable, but species within each group vary significantly in nesting materials, water contact, and incubation behavior. This undermines the strength of clustering conclusions based on nest type alone.

Thank you for this comment, which we agree with. We discuss this limitation in detail in detail in the Discussion (lines 381-388), concluding that “Future studies would benefit from including more wet-nesting species and more individuals per species. This will allow a more detailed and robust exploration of the determinants of fine-scale variation in bacterial abundance, such as types of nesting materials used, incubation behaviour and temporal influences on microbiota. Additionally, sampling more than one egg per clutch would be beneficial to account for intra-clutch variation in eggshell microbiota (e.g. due to egg laying order, partial incubation or brood parasitism), while using candling techniques to estimate embryo age within each egg would help account for microbial variation due to incubation stage.”

In addition, our concluding paragraph (lines 466-468) now states that “Future studies would benefit from increased sample sizes, both within and across species, to allow a robust exploration of not only nesting strategy on eggshell microbiota, but other ecological factors such as egg laying date, incubation, and nesting materials”.

Comment 3: One Egg per Nest Sampled**: Sampling only one egg per clutch does not account for intra-clutch variation, which can be substantial and could bias results - discuss this in the Limitations section.

Thank you for this comment. In lines 384-388 we now state that “Additionally, sampling more than one egg per clutch would be beneficial to account for intra-clutch variation in eggshell microbiota (e.g. due to egg laying order, partial incubation or brood parasitism), while using candling techniques to estimate embryo age within each egg would help account for microbial variation due to incubation stage.”

Comment 4: Bacterial communities change with incubation, and the study did not assess incubation stage or egg laying order. Without controlling for this, variability may mask true differences due to nesting strategy - discuss this in the Limitations section.

As mentioned above, we now acknowledge that a limitation of our study is our small sample sizes, which meant that we could not control for additional ecological factors such as incubation behaviour, brood parasitism, and temporal patterns, which indeed could mask true differences due to nesting strategy.

Comment 5: Many dry-nesting species have sample sizes of 1–3 nests, making any comparison across species or nesting strategies tenuous - discuss this in the Limitations section.

Yes, we agree this is a limitation of our study. We have slightly modified the limitations paragraph to now state that “Our sample sizes are higher for dry-nesters (eleven species), while we only include two wet-nesting species in our analysis, due to the limited number of wet-nesting species in the study area. In addition, samples sizes within species were often limited. This restricts the range of analyses that we can perform on our data, and could mask true differences in bacterial assemblages with nesting strategy” (lines 375-379).

Comment 6: Use of Outdated Genetic Method (ARISA). While ARISA is a valid fingerprinting technique, it has been largely supplanted by 16S rRNA sequencing and shotgun metagenomics, which offer finer resolution. The authors acknowledge this but should clearly frame ARISA's limitations.

We have now updated the description of ARISA to better frame its limitations, especially in the context of metagenomics and other more-modern techniques, stating that “Nonetheless, ARISA can still only provide an estimate of diversity due to issues associated with taxon-specific differences in genomic structure (e.g. some species do not have the 16S and 23S genes organised in an operon), leading to some bacteria being undetected and others over-represented….Despite recent advances in the molecular characterisation of bacterial communities, which allow finer resolution of bacterial assemblages (e.g. metagenomic techniques), ARISA and culture-based techniques remain effective tools to characterise broad differences in assemblages between treatment groups” (lines 136-145).

Comment 7: Avoid Overstating Conclusions. Many conclusions regarding the effect of nesting strategy are too strong given the limitations in design. These should be reworded to reflect the exploratory nature of the study.

We have reread our manuscript and softened our concluding statements, to better reflect the exploratory nature of the study.

Comment 8: Clarify Figures. Figures should include clearer legends and visual indicators (e.g., nesting strategy symbols or species groupings). Consider separating duck species from other dry-nesters in visualizations.

Thank you for this comment. We feel that the figure which could best use visual indicators for nesting strategies is Figure 2 (nMDS results for bacterial assemblage vs species/nesting strategy). This figure however already includes visual indicators (circles for dry nesters and triangles for wet-nesters). Although we like the idea of separating ducks from other dry-nesters in visualisations, we feel that this will overcomplicate the figure with too many symbol-types. Instead, the reader can identify each species by the colour of each symbol and by reading the key in the figure caption. We therefore prefer to leave the figure unmodified.

Comment 9: Specify that future research should explore egg antimicrobial traits, control for incubation stage, and incorporate 16S sequencing to identify microbial taxa involved in pathogenic vs. commensal roles.

We now conclude our Discussion by stating that “Future studies would benefit from increased sample sizes, both within and across species, to allow a robust exploration of not only nesting strategy on eggshell microbiota, but other ecological factors such as egg laying date, incubation, and nesting materials. Moreover, future research should adopt more advanced molecular approaches, such as 16S rRNA gene sequencing and metagenomics, to enable deeper and more accurate characterisation of microbial communities” (lines 466-471).

Comment 10: Line 23: Change “Animals host diverse bacterial assemblages” to “Birds host diverse bacterial assemblages” to reflect study focus.

We have made the suggested change.

Comment 11: Line 168: Add “the” before “media”: “…onto the media.”

We have made the suggested change.

Comment 12: Hyphenation: Standardize the use of “wet-nesting” and “dry-nesting” throughout the manuscript.

We have revised the consistent use of “wet-nesting” and “dry-nesting” throughout the manuscript. Broadly, we use the term "wet-nesting" as an adjectival descriptor of nesting strategy, while the noun phrase "wet nest" describes the physical structure itself. This distinction helps clarify whether we are referring to the actual nest or to the broader nesting strategy.

Comment 13: Tone Down Phrasing: Replace strong language such as “our findings are significant” with “our findings suggest” or “indicate.”

We have revised to manuscript to soften our conclusions throughout. In terms of your specific comment here, the statement we believe you are referring to is: “Our findings suggesting differences in bacterial load and diversity between wet-nesting and dry-nesting species are significant, as trans-shell penetration of bacteria can occur, which can increase embryonic mortality.” We have not modified this sentence as we state that our findings SUGGEST a difference and this possible difference WOULD be significant for the reasons described.

Comment 14: Line 196: Improve clarity of sampling description—consider a flowchart or table to summarize sample numbers.

In lines 116-121, we now state that “In total, we sampled 101 egg and 24 water samples (total 125 samples). All samples successfully produced cell cultures (i.e. total sample size of 125 samples). In contrast, the success of our DNA extraction for the genetic characterisation of microbiota was more variable, resulting in reduced sample sizes for these analyses. This corresponded to 73 egg and 22 water samples for all genetic analyses. See S2 Table for further details”.

Please note that, during our revision of sample sizes, we identified a minor error in the reported number for the across-egg similarity analyses (lines 336 and 341), which was initially stated as N = 82 (instead of N = 73). This discrepancy was limited to the reported value and did not affect the underlying analysis. To ensure consistency and accuracy, we reviewed sample sizes across all other analyses and confirmed that no further issues were present.

Comment 15: Phrase such as “highlight important differences” should be reworded to “suggest potential differences.”

Thank you. As stated above, we have now reread the manuscript and toned down these strong statements.

Comment 16: Ensure consistent italicization of all Latin species names throughout the manuscript, including the References.

We have revised the manuscript and made any relevant changes. Please note that ‘coliforms’ is not a genus (or species) name and is therefore no italicised.

Comment 17: The manuscript references older papers for methods but does not cite newer alternatives (e.g., 16S sequencing), which should at least be discussed.

We now mention metagenomic techniques in the Methods (line 143), and also provide several references to more modern genetic techniques in the Discussion (lines 469-470).

Reviewer 3

Comment 1: Why you changed corresponding author in the revised version of the manuscript and why one of the co-authors (L. Rubáčová) has been removed from the author team to Acknowledgements only? Did she agree with this and it was discussed with her thoroughly? It is not usual such dramatically change author membership and status of authors?

The corresponding author was changed in consultation with PloSOne and was a necessity to allow for payment for the publication of this manuscript. Despite this change, all co-authors were involved in the revisions of the manuscripts.

L. Rubáčová contributed to the acquisition of the data used in this manuscript. While we initially included her as a co-author in recognition of this contribution, she later expressed a preference to be acknowledged rather than listed as an author, feeling that her involvement did not meet her personal threshold for authorship. We respect and appreciate her perspective and have honoured her request by acknowledging her contribution accordingly.

Comment 2: To your comments regarding clutch age (my (Reviewer 3) Comment 1) and effect on eggshell bacterial assemblage. Yes, you considered sampling date and describe that effect might be linked with changed environmental factors. But, I am considering incubation of the clutch and resulting eggshell assemblage. As you described on lines 150-152, you sampled one egg from nest when you found it, but although you record the precise date of nest finding, you were not aware of the age of the clutch (i.e. how long the clutch has been incubated already). Moreover, it is quite common that 1) nesting parasitism occurred in Mallards and also coot and grebes and also that 2) intermittent incubation of partial not-completed clutch occurred before the full incubation and clutch completion (at least in ducks it is very common, and I think also swans or grebes are known for this behaviour). In practice, simple fieldwork “candler” (inspection of egg inner structure under the light) is used for determination of embryo age which correspond to intensity of incubation. And as you mentioned in you text of the revised ms that incubation may affect eggshell bacterial assemblage (and quite a lot), this discrepancy in eggs age and developmental stage within the clutch may cause bias if your chosen only one egg from the complete clutch for sampling. I think you should somehow touch of this possible bias in your data caused by the different clutch age, role of intermittent incubation and/or brood parasitism in your focal species.

Thank you for this further clarification. We have now discussed this in more detail in lines 384-388, considering the reviewer’s comments, stating that “Additionally, sampling more than one egg per clutch would be beneficial to account for intra-clutch variation in eggshell microbiota (e.g. due to egg laying order, partial incubation or brood parasitism), while using candling techniques to estimate embryo age within each egg would help account for microbial variation due to incubation stage.”

Comment 3: But in general, it is nice you added additional GLMMs testing for the effect of clutch size and sampling date, as it is really highly probably that enlarged clutches may be incubated before clutch completion or being parasitized by brood parasites. This additional GLMMs made your conclusions more robust. Yet, still, as your study is pioneering in study of bacteria on eggshell with respect to breeding ecology, I am for to include possible factors that might be behind the differences between dry- and wet-nesters. Although I understand your point of view and you want to keep the line of the study as simple as possible, I think a few sentences in the discussion with other potential factors cannot be distracting from the major message of your study.

Thank you. As mentioned above, we have now added discussion of other possible factors behind the differences in wet- and dry-nesters (e.g. lines 381-388 and 407-416).

Comment 4: Line 141 - “more holistic overview”., I suggest to replace holistic by “comprehensive”. Holistic is really to strong meaning in your case study.

Suggested change made.

Comment 5: Line 142 - …in isolation. I suggest to replace by “separately” or some other equivalent.

Suggested change made.

Comment 6: Line 286-287 – “….of bacteria on eggshells on wet-nesting..“ rephrase it please.

Suggested change made

Comment 7: Line 296 – “coliforms” without capital please

Suggested change made

Comment 8: Lines 359-366 – I am still wondering if it cannot be partially caused by pre-incubation behaviour of dry-nesters – covering of clutch by down feathers in mallard and other duck species, intermittent incubation I have mentioned above. I think you should take this into account in discussion of your results (mostly differences between dry and wet-nesters). You already did for covering the clutch by feathers, and even if you disagree to do for incubation (see response to my Comments 18). I think it can help to untangle the all possible confounding factors that have to be taken into account in future studies of eggshell or trans-shell bacterial assemblages.

Thank you for this comment. As explained above, we have now acknowledged the potential role of factors such as partial incubation and brood parasitism on differences between wet- and dry-nesters (lines 384-388). We also stated that “Finally, some species may display intermittent incubation or experience brood parasitism, which can further complicate interspecies explorati

---

## [Editor Report · Decision Letter 2]

30 Jul 2025

Dear Dr. Brandl,

Thank you for submitting your manuscript to PLOS ONE. After careful consideration, we feel that it has merit but does not fully meet PLOS ONE’s publication criteria as it currently stands. Therefore, we invite you to submit a revised version of the manuscript that addresses the points raised during the review process.

We look forward to receiving your revised manuscript.

Kind regards,

Petr Heneberg

Academic Editor

PLOS ONE

**Journal Requirements:**

**Additional Editor Comments:**

The manuscript has been substantially improved after the previous round of revisions and now covers most of the points that were previously raised. However, there are still a number of issues that should be addressed before it is ready for publication. These remaining points concern interpretation, precision of wording and clarity of figures and data presentation rather than additional data collection.

The conclusions still extend beyond what the data can support. Several sections of the abstract and discussion state or imply that wet-nesting results in higher bacterial loads and that this has consequences for embryonic mortality. This study is observational and does not test cause and effect or hatching outcomes. These implications should be presented as hypotheses for future research rather than results supported by the current data. In particular, the discussion around lines 443 to 444 and the associated summary sentences in the abstract need to be reworded to reduce the strength of the claims.

The categorization of wet-nesting versus dry-nesting species is too broad. This point was acknowledged but the language in the results and discussion still reads as if general conclusions across these categories can be made. The sample of wet-nesting species is restricted to two closely related species and the dry-nesting group is heterogeneous in water contact and nest materials. Each time the results are generalized across these categories, an explicit caveat about these limitations should be included.

The possibility that phylogeny influences bacterial clustering has not been addressed with sufficient emphasis. Both wet-nesting species belong to the same family. Clustering patterns could therefore reflect shared evolutionary history rather than nest characteristics. The discussion should explicitly highlight that these factors cannot be separated in the present dataset and that the effect of nesting strategy should be interpreted with caution.

The clarity of the figures remains an issue. Figure 2 in particular is difficult to interpret due to overlapping symbols. Although the legend provides color codes, the visual separation of species is poor. Symbols could be made partially transparent, larger, or shown in separate panels for wet and dry nesters. The figure caption should also report the stress values for the multidimensional scaling. These were previously requested but have not been implemented.

The description of patterns in bacterial similarity between eggs and water could be misinterpreted. The text attributes differences mainly to ecological filtering, yet differences in the sampling methods and the inherent biases of ARISA are also plausible explanations. These limitations should be clearly distinguished from ecological interpretations.

The study relies on ARISA and culture-based methods, and while this is acceptable for exploratory work, these approaches provide low resolution. The methods section and the discussion mention these limitations, but they should be restated more clearly in the abstract and conclusion. This would help to place the findings in the context of recent research that uses higher resolution molecular approaches.

Finally, one of the earlier reviewer requests to visually separate ducks from other dry-nesting species in the figures was declined by the authors. If this decision is retained, the text should guide the reader through the interpretation of symbol codes more explicitly so that the influence of ducks on clustering patterns can be evaluated by readers.

In summary, the manuscript contributes useful new data on eggshell-associated bacteria in wetland birds. It now acknowledges many of the limitations of the study but still needs further softening of causal language, an explicit discussion of phylogenetic influences, clearer presentation of figures and more careful integration of limitations into the conclusions. Addressing these points will strengthen the work and ensure that the results are interpreted within the constraints of the study design.

---

## [Author Response · Author response to Decision Letter 3]

28 Aug 2025

Academic Editor

Comment 1: The conclusions still extend beyond what the data can support. Several sections of the abstract and discussion state or imply that wet-nesting results in higher bacterial loads and that this has consequences for embryonic mortality. This study is observational and does not test cause and effect or hatching outcomes. These implications should be presented as hypotheses for future research rather than results supported by the current data. In particular, the discussion around lines 443 to 444 and the associated summary sentences in the abstract need to be reworded to reduce the strength of the claims.

Response: We have rewritten sections of the Abstract and Discussion to present our interpretations more cautiously. These changes highlight that our data suggest an intriguing link between nesting strategy and eggshell-associated bacteria, but that more comprehensive sampling, with more advanced genetic techniques, are required to confirm these patterns (e.g. lines 39-42, lines 368-371 and 463-474).

In addition, we have greatly reduced our discussion on trans-shell penetration in the Discussion section. We agree that it is too early to make detailed speculation on the consequence of nesting strategy in the study species on embryonic survival. We now only briefly mention this as a cautious possibility in the penultimate paragraph of the Discussion (lines 455-462)

Comment 2: The categorization of wet-nesting versus dry-nesting species is too broad. This point was acknowledged but the language in the results and discussion still reads as if general conclusions across these categories can be made. The sample of wet-nesting species is restricted to two closely related species and the dry-nesting group is heterogeneous in water contact and nest materials. Each time the results are generalized across these categories, an explicit caveat about these limitations should be included.

(In addition), the possibility that phylogeny influences bacterial clustering has not been addressed with sufficient emphasis. Both wet-nesting species belong to the same family. Clustering patterns could therefore reflect shared evolutionary history rather than nest characteristics. The discussion should explicitly highlight that these factors cannot be separated in the present dataset and that the effect of nesting strategy should be interpreted with caution.

Response: We have now revised our text to better clarify the limitations of our data, in particular the close-relatedness of the two wet-nesting species and the broad phylogeny of the dry-nesting species. We now consistently state that the wet-nesting species are closely-related, while our category of dry-nesting is broad (e.g. lines 106-111, lines 361-364 and 410-412). The paragraph in lines 410-432 continues to discuss the fact that the study requires an increase in species in both categories to make definitive conclusions on nesting strategy and eggshell bacteria. We also discuss the fact that the dry-nesting category is very broad and includes species with very different ecologies and behaviours, which further complicates the interpretation of our data. We have added a sentence stating that “our nMDS analyses suggested no obvious clustering of bacterial assemblages of the family Anatidae, which includes ducks, geese and swans, despite their unique nesting behaviours (e.g. intermittent incubation)” (lines 421-423). This latter point is illustrated in Figure 2, where we have updated the graph symbols to clearly indicate which species belong to the Anatidae family.

Comment 3: The clarity of the figures remains an issue. Figure 2 in particular is difficult to interpret due to overlapping symbols. Although the legend provides color codes, the visual separation of species is poor. Symbols could be made partially transparent, larger, or shown in separate panels for wet and dry nesters. The figure caption should also report the stress values for the multidimensional scaling. These were previously requested but have not been implemented.

(In addition), one of the earlier reviewer requests to visually separate ducks from other dry-nesting species in the figures was declined by the authors. If this decision is retained, the text should guide the reader through the interpretation of symbol codes more explicitly so that the influence of ducks on clustering patterns can be evaluated by readers.

Response: We have revised Figure 2 in response to concerns about its interpretability. The updated figure features larger, semi-transparent, and more distinct, symbols to enhance visual differentiation among species. Additionally, a visual legend has been added at the bottom of the figure to clearly associate each symbol with its corresponding species.

Members of the Anatidae family (swans, ducks, and geese) are now highlighted using open circles. This modification more clearly demonstrates a likely lack of clustering of bacterial assemblages of the family Anatidae, despite their unique nesting behaviours. This is now also mentioned in the Discussion (lines 421-423).

Finally, we have added the stress value for the nMDS in the figure caption, which was unintentionally missed in previous revisions of this manuscript. We hope these modifications improve the clarity and usefulness of the figure.

Comment 4: The description of patterns in bacterial similarity between eggs and water could be misinterpreted. The text attributes differences mainly to ecological filtering, yet differences in the sampling methods and the inherent biases of ARISA are also plausible explanations. These limitations should be clearly distinguished from ecological interpretations.

Response: Yes, we agree that the water data should be interpreted with caution. As noted in lines 274–282 of the Methods, the differing approaches used to sample eggshells and water constrain the conclusions that can be drawn when comparing egg-associated and water-associated bacteria. Accordingly, we have already limited the scope of our analyses involving water-associated bacteria.

In addition, we have now modified the text in the Discussion on the limitations of the analyses with the water-associated bacteria. We stress that, as the Bray-Curtis dissimilarity index used in the nMDS incorporates both OTU identity and abundance, field and genetic sampling issues may obscure true variation in bacterial assemblages associated with nesting strategy (lines 433-444). We then continue by discussing the possible ecological explanations for the observed dissimilarity between eggshell-associated and water-associated bacteria.

Comment 5: The study relies on ARISA and culture-based methods, and while this is acceptable for exploratory work, these approaches provide low resolution. The methods section and the discussion mention these limitations, but they should be restated more clearly in the abstract and conclusion. This would help to place the findings in the context of recent research that uses higher resolution molecular approaches.

Response: We have now added additional text to both the abstract and conclusion to highlight that, while our results are promising, further research with more advanced molecular approaches are required. Specifically, in the Abstract we state “Further research is, however, required to confirm these patterns, incorporating more comprehensive sampling and utilising more advanced genetic approaches” (Lines 41-42). In addition, the Conclusion now states that “future research should adopt more advanced molecular approaches, such as 16S rRNA gene sequencing and metagenomics, to enable a deeper and more accurate characterisation of microbial communities” (lines 472-474).

---

## [Editor Report · Decision Letter 3]

31 Aug 2025

Bacterial assemblages on eggs reflect nesting strategies in wetland-associated birds

PONE-D-24-57432R3

Dear Dr. Brandl,

We’re pleased to inform you that your manuscript has been judged scientifically suitable for publication and will be formally accepted for publication once it meets all outstanding technical requirements.

Kind regards,

Petr Heneberg

Academic Editor

PLOS ONE
---

## [Editor Report · Acceptance letter]

PONE-D-24-57432R3

PLOS ONE

Dear Dr. Brandl,

I'm pleased to inform you that your manuscript has been deemed suitable for publication in PLOS ONE. Congratulations! Your manuscript is now being handed over to our production team.

Kind regards,

on behalf of

Dr. Petr Heneberg

Academic Editor

PLOS ONE